# Optical Coherence Tomography as a Valuable Tool for the Evaluation of Cutaneous Kaposi Sarcoma Treated with Imiquimod 5% Cream

**DOI:** 10.3390/diagnostics13182901

**Published:** 2023-09-11

**Authors:** Carmen Cantisani, Alexandru-Vasile Baja, Luca Gargano, Giovanni Rossi, Marco Ardigò, Giuseppe Soda, Mehdi Boostani, Norbert Kiss, Giovanni Pellacani

**Affiliations:** 1Dermatology Unit, Department of Clinical Internal Anesthesiological and Cardiovascular Sciences, “Sapienza” University of Rome, 00161 Rome, Italy; alexandru.baja@gmail.com (A.-V.B.); lucagargano1995@gmail.com (L.G.); giovanni.rossi@uniroma1.it (G.R.); giuseppe.soda@uniroma1.it (G.S.); giovanni.pellacani@uniroma1.it (G.P.); 2Porphyria and Rare Diseases Unit, San Gallicano Dermatological Institute—IRCCS, 00144 Rome, Italy; ardigo.marco@gmail.com; 3Department of Dermatology, Venereology and Dermatooncology, Semmelweis University, 1085 Budapest, Hungary; mehdi_parsii@yahoo.com (M.B.); kiss.norbert@med.semmelweis-univ.hu (N.K.)

**Keywords:** kaposi sarcoma, optical coherence tomography, imiquimod, topical treatment, dermoscopy, imaging

## Abstract

Kaposi sarcoma (KS) is a rare disease that was not frequently identified before the widespread occurrence of AIDS. Even today, it remains a challenge for physicians to diagnose, particularly in its early stages, often requiring referral to specialists and further investigations. Dermoscopy, a non-invasive imaging technique, reveals a distinctive rainbow pattern that strongly indicates KS. Moreover, advanced imaging tools like optical coherence tomography (OCT) can provide additional information though specific disease-related patterns have not been fully established yet. These emerging techniques show promise in facilitating early diagnosis of skin-related KS and monitoring the effectiveness of treatments. However, biopsy remains the definitive method for confirming the disease. In this study, we present two cases of cutaneous Kaposi sarcoma, documented using OCT, both before and after treatment with imiquimod 5% cream. The study highlights the potential of OCT in evaluating disease progression and treatment response, as well as the usefulness of dermoscopy in detecting early indicators of KS. By integrating these advanced imaging techniques, the diagnosis and management of cutaneous KS could be improved, leading to timely interventions and better patient outcomes.

## 1. Introduction

Kaposi sarcoma (KS) is considered a low-grade vascular tumor, induced by human herpesvirus-8 (HHV-8), also known as Kaposi’s sarcoma-associated herpesvirus (KSHV). KS predominantly affects mucocutaneous sites but may involve all organs [1].

KS has four distinct clinical and epidemiological forms: classic, endemic (common in African regions), epidemic (associated with HIV-positive patients), and iatrogenic (develops in immunosuppressed individuals). Classic KS primarily affects elderly men and is more prevalent among people from Mediterranean regions. This vascular tumor is typically found on the lower extremities, particularly the ankles and feet. However, in individuals with AIDS-related lesions, the trunk is frequently affected [2,3].

In addition to HHV-8 [4], other contributing factors, such as cytokine-induced growth, have been associated with the development of KS and other diseases like primary effusion lymphoma and multicentric Castleman disease [5,6]. While AIDS-related KS and iatrogenic KS are clearly linked to immunosuppressive conditions, classic KS may involve immunostimulation or immune dysregulation in its pathogenesis. The exact cause has not been fully determined, but HHV-8 infection is thought to potentially initiate the processes [7]. Endemic KS, unlike AIDS-related and iatrogenic KS, has not shown conclusive evidence of underlying immunodeficiency [8].

In affluent regions such as the U.S. and Europe, suspected cases of KS are routinely confirmed through histopathology. Given that the skin lesions are typically visible, obtaining tissue for microscopic confirmation is easily achieved with a simple skin punch biopsy. However, in sub-Saharan Africa, where KS is clinically relevant, the diagnosis is often based on visual inspection of the lesions without histologic confirmation [9].

The infectious cause of all types of KS (classic/sporadic, iatrogenic/posttransplant, endemic/African, and AIDS-related/HIV-associated) has been identified as HHV-8 [4]. HHV-8 expresses the latent nuclear antigen-1 (LAN-1) in all infected cells [10]. Immunohistochemically identifying HHV-8 through LAN-1 has become the gold standard for diagnosing KS and distinguishing it from similar conditions. This method has demonstrated high specificity and sensitivity in diagnosing KS [11,12,13,14].

Several technologies have been developed as novel non-invasive diagnostic methods, including dermoscopy [15], reflectance confocal microscopy (RCM) [16], and optical coherence tomography (OCT) [17].

Dermoscopy is a technique commonly used to examine skin tumors and other skin conditions [15,18]. It operates on the principles of epiluminescence microscopy, where the dermatoscope’s optical system enables the visualization of skin structures beneath the surface. By illuminating the skin with either polarized or non-polarized light, dermoscopy enhances the visualization of subsurface structures, pigmentation, and vascular patterns that are not readily observable with the naked eye. The magnification capabilities of the dermatoscope allow clinicians to observe intricate details and differentiate between benign and malignant lesions based on characteristic features unique to various skin conditions [19]. In 2009, Hu and colleagues initially documented the dermoscopic characteristics of KS. The primary observed dermoscopic features included bluish–reddish coloration, a rainbow pattern, a scaly surface, and small brown globules [3].

RCM is an advanced imaging method that offers non-invasive, high-resolution visualization of the epidermis and papillary dermis at the cellular level. Utilizing the distinctive contrast offered by melanin and melanosomes in RCM images, this technique facilitates a microscopic analysis of tissue morphology in both healthy skin and melanocytic lesions, effectively providing a “virtual” skin biopsy with detailed en-face views of various skin layers [16]. In the context of KS, three specific features exclusively evident at RCM examination were identified, described, and considered: disappearance of the normal papillary ring structures at the dermoepidermal junction level named non-rimmed papillae; disarrangement of dermal normal structures with a collection of newborn vessels, spindle, and inflammatory cells named dermal granulation-like tissue; and ill-defined fusiform, linear, or fibrillar crossing bands with stronger brightness than normal collagen and elastic fibers named irregular net-like fascicles [20].

OCT holds tremendous potential in the field of skin cancer detection, offering a promising pathway for future advancements. Initially developed in the late 1980s for ophthalmology applications [21,22], OCT is an imaging technique that utilizes laser technology, Michelson interferometry, and infrared light [17] to produce cross-sectional images of tissue by capturing backscattered light [23]. This process involves dividing the light from an optical source into two paths [24], where one is directed towards the tissue sample and the other towards a reference mirror [17]. Subsequently, when the backscattered photons from the skin recombine with the reference signal [24], an interference signal emerges if the lengths of the two paths align within the coherence length [17].

No universally accepted staging classification exists for KS. However, patient management considers three clinical scenarios: localized non-aggressive, locally aggressive, and disseminated KS [25]. Local treatments are effective for limited forms, while a symptomatic, palliative approach is suitable for extensive and progressive disease. Radiotherapy proves highly efficient for localized KS. Surgical excision carries a high recurrence rate but may be considered for specific well-defined limited and superficial lesions. CO2-laser and superficial cryotherapy are viable alternatives [25].

Imiquimod functions as an immune response modulator, operating via toll-like receptor 7, thereby instigating innate and adaptive immune responses. Importantly, it lacks direct antiviral efficacy but rather elicits the upregulation of interferon alpha genes, alongside interleukin 6 and 8 (IL-6 and IL-8) [26]. In a prospective, single-center, unblinded phase II trial, aimed at skin lesions of classic or endemic KS, the intervention exhibited antitumor efficacy in nearly half of the 17 participants. Concurrently, localized pruritus and erythema were documented in 53% of the patients [27]. Within the context of our clinical practice, we present two cases of localized cutaneous KS, wherein the application of OCT proved exceptionally valuable for lesion characterization, particularly when assessing topical treatment such as imiquimod 5% cream. Our analysis employed a Vivosight^®^ device equipped with advanced high-definition (HD) OCT processing software version 4.1 (Michelson Diagnostics Ltd., Maidstone, Kent, UK).

## 2. Case Presentation

### 2.1. Case Report 1

We present a case involving a 49-year-old African American man with a medical history that includes disseminated endemic KS (previously treated with doxorubicin), pulmonary tuberculosis, and pulmonary aspergilloma. During a period of systemic remission, the patient developed discrete violaceous papules on his right leg (Figure 1). The patient’s dark skin tone (phototype VI) added complexity to the diagnostic process, as it is well known that dermoscopic patterns can vary based on skin tone.

Dermoscopy revealed faint nodular structures on a red whitish background (Figure 2a). OCT was utilized to evaluate the lesions (Figure 2b–d). Lesional vascularization was assessed by measuring diameter and analyzing branching patterns, following established OCT standards.

The lesion was effectively managed by applying imiquimod 5% cream once daily for five days, with a total treatment duration of six weeks (Figure 3a–c). Post-treatment, the patient experienced only mild residual inflammation. Regular follow-up in the clinic continues for the patient.

### 2.2. Case Report 2

In our second case, we present a 60-year-old woman who reported multiple pinkish papules on her left ankle that had been present for the past four months. Her medical history includes a lung transplant performed approximately ten years ago, and she has been on continuous immunosuppression regimens ever since.

The clinical examination and dermoscopic analysis were conducted before (Figure 4a,b) and after (Figure 5a,b) the successful treatment with imiquimod 5% cream. The cream was applied once daily for five days a week, over a six-week treatment period. The visual documentation clearly demonstrates the effectiveness of the treatment, as the papules showed significant improvement after the course of therapy. Dermoscopy confirmed the reduction of prominent violaceous lesions, with only residual inflammation evident post-treatment.

To gain further insights into the lesion’s characteristics, pre- and post-treatment OCT scans were performed (Figure 4d–f and Figure 5c–e). These scans provided valuable information about the lesion’s structure and vascularity, offering a comprehensive view of its response to treatment.

Upon histological examination, plump sheets of spindle cells compressing the vasculature were notably present (Figure 4c), which supported the diagnosis.

The successful management of the pinkish papules on the patient’s ankle using imiquimod 5% cream showcases its efficacy in treating such lesions, even in individuals with complex medical histories, such as the patient who had undergone a lung transplant with long-term immunosuppressive therapy.

Regular monitoring and follow-up will continue to be an essential part of the patient’s care to ensure the long-term efficacy and safety of the treatment plan.

## 3. Discussion

The purpose of presenting these two cases of KS is to underscore the potential significance of non-invasive diagnostic techniques, particularly OCT, in effectively monitoring the response of KS to topical therapies. As previously mentioned, it is essential to note that a universally accepted staging system for KS is yet to be established.

Both of the cases we have reported are characterized by localized KS, involving limited and non-rapidly progressive lesions in the lower extremities. In this specific clinical scenario, there exists a diverse array of local therapeutic options, encompassing both destructive and pharmacological approaches [28].

However, these two cases serve to elucidate how the concept of localized disease must be contextualized in the clinical setting. In the first case, the localized KS represents a limited recurrence of a previously systemic disease, while in the second case, it denotes the initial onset of the disease. For both patients, topical therapy with imiquimod was selected, given that the lesions were only slightly elevated. The existing literature on the use of this topical immunomodulator in KS includes some case reports, one prospective phase II cohort study, and one comparative single-blinded non-controlled clinical study; however, these studies are insufficient to define the precise regimen dosage scheme [27,29].

In both cases, OCT was utilized for assessing the lesions during the diagnostic stage and subsequently to evaluate the response to therapy. While OCT has primarily been studied in non-melanoma skin cancers, it is being explored for new indications. In the context of KS, the current OCT examination reveals features that distinguish it from actinic keratosis (AK) and in situ squamous cell carcinoma.

Notably, KS exhibits a nodular, inhomogeneous area with signal attenuation in the dermis, accompanied by vascular-like signal-poor structures [30]. Our case suggests that certain parameters can be considered at the diagnostic stage before resorting to biopsy. To date, very few cases of KS with OCT evaluation have been reported in the literature, but those that have demonstrated atypical vascularization characterized by thin horizontally disposed vessels [31]. This vascular pattern, with organized thin vessels in the early stages, appears to be a distinct characteristic. Furthermore, this pattern corresponds with what is observed in histology, where early-stage KS (flat-papular forms) exhibits irregular, slit-like vascular spaces with thin walls and irregular morphology, along with the presence of a perivascular mononuclear infiltrate containing lymphocytes and plasma cells.

In contrast, AK and in situ squamous cell carcinoma are characterized by irregular epidermal thickening and hyperkeratosis. Furthermore, the OCT examination can differentiate KS from basal cell carcinoma (BCC), as the latter typically exhibits lobular tumoral structures in the dermis [30].

Recently, a case report detailed the use of OCT in extramammary Paget’s disease. Before radiation therapy, Low-Coherence OCT was conducted, revealing specific characteristics of Paget cells, which were larger than keratinocytes, displaying a dark cytoplasm and a slightly bright nucleus, and organized in nest-like structures. Following radiotherapy, both clinical examination and OCT confirmed a complete response to treatment. Additionally, the pre-treatment OCT findings have potential applicability in monitoring any sub-clinical residual disease [32].

In the context of BCC, OCT shows promise in assessing treatment efficacy, particularly with Imiquimod therapy, aiding in detecting residual disease within healed tumors [33]. OCT is also useful for assessing lesion margins and depth, assisting in treatment decisions between topical and surgical options [34]. Additionally, OCT correlates well with biopsy depth measurements, making it a valuable tool for pre-treatment assessment, especially for BCCs less than 4 mm deep. Similar considerations apply to KS for selecting suitable neoplasms for imiquimod therapy [35].

In another study by Forsea et al. [30], preliminary results were presented from a study focusing on the diagnostic potential of OCT in epithelial cancers and precancers, including Kaposi’s angiosarcoma. They included 15 consecutive patients with suspected epithelial lesions and compared OCT evaluations with clinical digital photography and contact dermoscopy. Combining OCT with dermoscopic evaluation improved the diagnostic performance compared to clinical assessment alone or using OCT or dermoscopy individually. The study also confirmed specific diagnoses for the participants, with notable findings of Kaposi’s angiosarcoma cases.

In the study conducted by Cappilli et al. [36], distinct line-field confocal optical coherence tomography (LC-OCT) features were identified for various cutaneous lesions, significantly enhancing their differential diagnosis from KS. Specifically, in the case of angiokeratomas, vertical LC-OCT images revealed a hyperkeratotic epidermis with corneum disruption and irregular acanthosis (12/15, 80.0%). For all angiokeratomas, large oval dark spaces assuming a lobular arrangement occupied the whole visible dermis both in vertical and horizontal LC-OCT images, distinguishing them from KS. Pyogenic granulomas exhibited numerous small linear and circular dark regions within the upper dermis (6/6, 100%), as well as bright spots in the upper dermis (6/6, 100%), along with expanded inter-keratinocyte spaces in the epidermis (5/6, 83.3%) as visualized by LC-OCT. Thrombosed hemangiomas displayed multiple dark oval-shaped regions with varying degrees of brightness and amorphous content, enclosed by well-defined bright septa, and established a lobular pattern in all assessed lesions when viewed through vertical, horizontal, and 3D LC-OCT perspectives. Venous lakes were characterized by the presence of featureless regions (4/5, 80%) and pale structures (2/4, 50%), disclosing a substantial circular depth (5/5, 100%) in LC-OCT, in conjunction with illuminated connective tissue (4/5, 80%), respectively. Additionally, targetoid haemosiderotic hemangiomas LC-OCT images exhibited linear/round regions of darkness and well-defined bright spots within the dermal layer (4/4, 100%), respectively [36].

In contrast, KS exhibited highly distinctive LC-OCT features, specifically linear dark regions within the upper dermis (4/4, 100%) and dark intersections involving multiple irregular vessels. This specific LC-OCT pattern effectively differentiates KS from other vascular lesions. The incorporation of these OCT features into the diagnostic process holds significant potential for enhancing our ability to discriminate KS from its mimicking conditions, thus contributing to more precise identification and improved management of this complex disease [36].

Finally, in both of our cases, follow-up OCT examinations demonstrated a reduction in vascularization after therapy, with notable decreases in tortuosity and vessel thickness. These findings underscore the potential significance of OCT parameters in assessing the response to topical therapy.

## 4. Conclusions

Our case reports offer a comprehensive longitudinal assessment of KS lesions using dermoscopy and OCT. Moreover, they reinforce the appropriateness and safety of imiquimod as a viable treatment option for localized forms of KS.

## Figures and Tables

**Figure 1 diagnostics-13-02901-f001:**
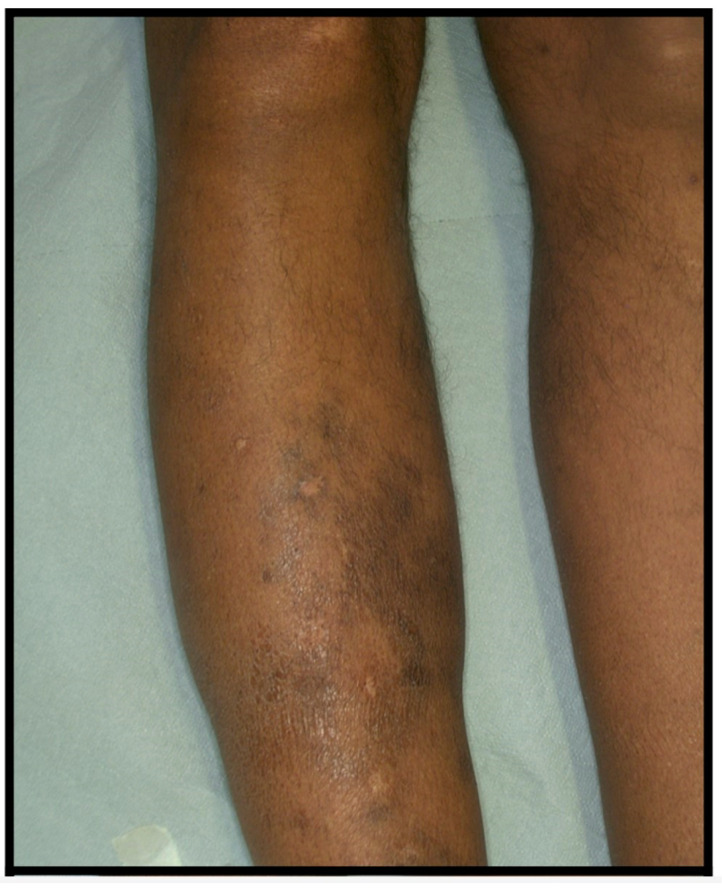
Multiple brownish-livid superficial plaques on the right shin.

**Figure 2 diagnostics-13-02901-f002:**
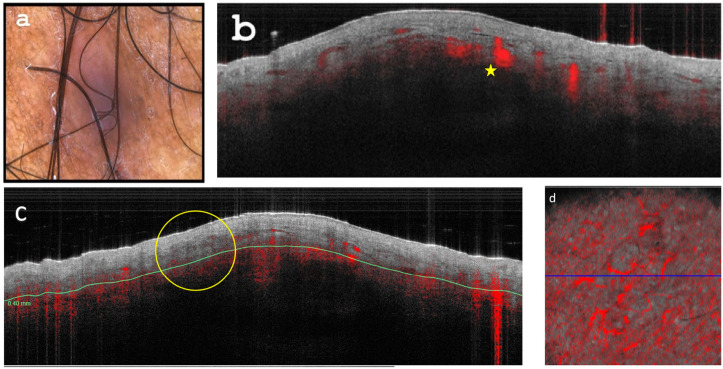
(**a**) Dermoscopy depicts a blue-violaceous papular lesion exhibiting peripheral fading and a white-shiny glare, consistent with anisotrichosis. (**b**) Optical Coherence Tomography (OCT) cross-sectional capture reveals a nodular area with dermal attenuation, accompanied by visibly enlarged lesional vascularization (indicated by yellow stars). (**c**) Tissue inhomogeneity visible on OCT (yellow circle). (**d**) Enface OCT image: Demonstrating enhanced, branching vascular pattern (64 qm diameter) at a depth of 0.3 mm.

**Figure 3 diagnostics-13-02901-f003:**
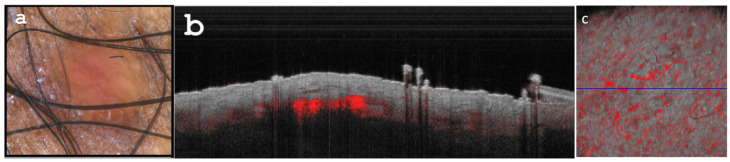
(**a**) Dermoscopy reveals the flattening of the lesion with persistent, yet reduced inflammation. (**b**) Optical coherence tomography (OCT) B scan (cross-sectional capture) displays diminished dermal attenuation and a scarce, thinned vascular pattern. (**c**) Enface OCT image showcases diminished vascularization (with a diameter of 40 qm) obtained at a depth of 0.3 mm, following topical imiquimod treatment.

**Figure 4 diagnostics-13-02901-f004:**
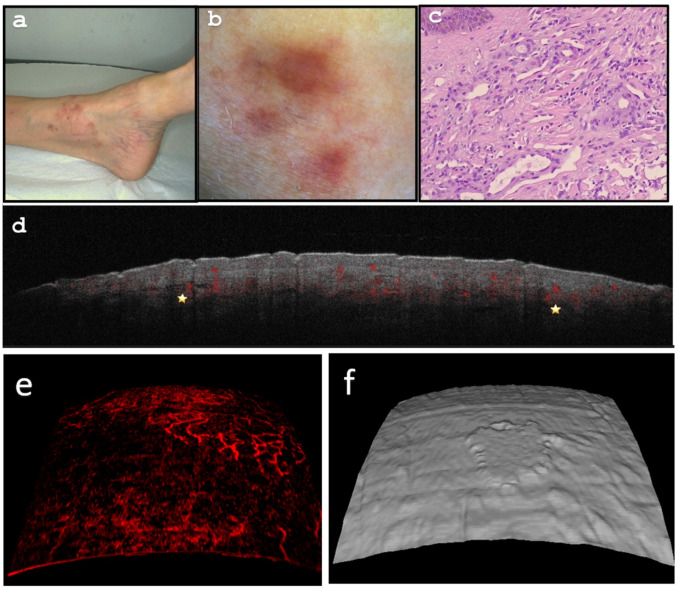
(**a**) Multiple superficial, coalescing livid-erythematous papules located on the left ankle. (**b**) Dermoscopy presents homogenous livid-erythematous papules with peripheral fading. No rainbow pattern is evident. (**c**) Histology (Hematoxylin and Eosin stain, ×100 magnification) exhibits plump sheets of spindle cells compressing blood vessels. (**d**) Optical Coherence Tomography (OCT) B scan (cross-sectional capture) reveals a dermal attenuation pattern with increased vascularization (indicated by yellow stars). (**e**) Enface OCT image at 0.3 mm depth exhibits enhanced, mesh-like vascularization with noticeable tortuosity (diameter approximately 60 qm). (**f**) OCT 3-dimensional lesional reconstruction displays raised papulo-nodular characteristics.

**Figure 5 diagnostics-13-02901-f005:**
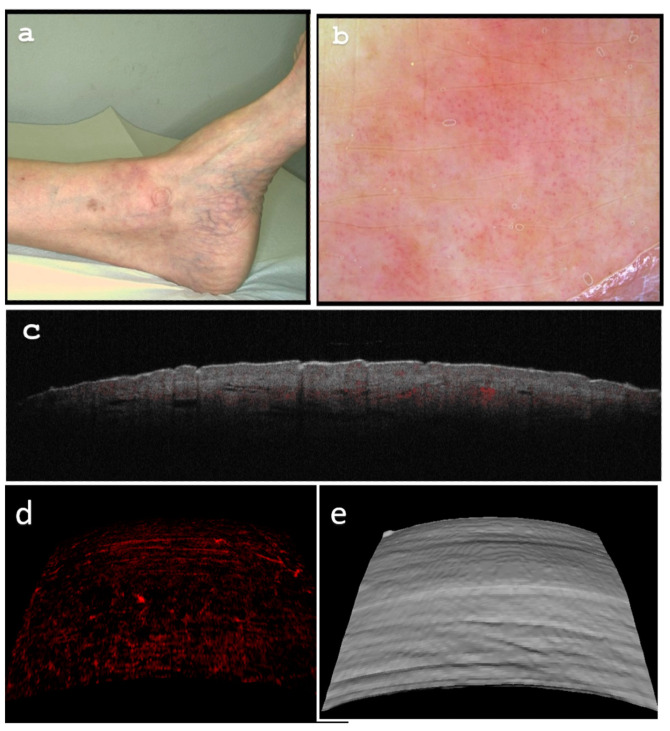
(**a**) Multiple faint livid-pinkish residual papules. (**b**) Dermoscopy reveals lesional regression on a background of mild residual inflammation with a pinpoint vascular pattern. (**c**) On Optical Coherence Tomography (OCT) B scan (cross-sectional capture), residual vessel thinning is observed with diminished dermo-epidermal attenuation. (**d**) Enface OCT image (at 0.3 mm depth) showcases diminished post-treatment lesional vascularization (with a diameter of 35 qm). (**e**) OCT 3-dimensional lesional reconstruction of the lesion illustrates a flattened and confluent appearance with the surrounding dermis.

## Data Availability

The data supporting the study findings are available from the corresponding author C.C. upon reasonable request.

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
