# Peer review of "Optical Coherence Tomography as a Valuable Tool for the Evaluation of Cutaneous Kaposi Sarcoma Treated with Imiquimod 5% Cream"

_diagnostics, 2023, doi:10.3390/diagnostics13182901_

Round 1

Reviewer 1 Report

1. In lines 110, 208, and 223 instead of KS you say "KS sarcoma"

2. For patient 2 you say "a pinkish papule on her left ankle" but in figure 4a multiple papules are visible. Which one is correct? If the papule is solitary (in yours or other cases) doesn't removal with biopsy affects the after-treatment OCT image?

3. Are OCT characteristics of KS (described in Discussion) specific for the disease? Is there any knowlegde/literature about the OCT characteristics of clinically similar diseases, like bacillary angiomatosis, arteriovenous malformations, cutaneous lymphoma etc? A paragraph in Discussion about differential diagnosis using OCT would be useful for clinicians reading the article.

4. On the other hand, in Discussion again, extensive analysis of OCT for BCC seems a bit out of place. It is not a review about differentiating skin cancers with OCT. This part could be a little smaller or maybe add information about diseases clinically similar to KS

Author Response

We appreciate the valuable feedback from the reviewer, and as a result, we have made the following corrections:

1-We have addressed the inconsistency regarding the usage of "KS sarcoma" and have corrected it to "KS."

2-The reviewer pointed out the inconsistency in the description of Figure 4a, where "multiple papules" is the correct description. We have rectified this inconsistency. Furthermore, the site of biopsy was not assessed by OCT. 

3-We concur with the reviewer's suggestion to enhance the paper's quality by adding literature references for other cutaneous lesions, aiding clinicians in differential diagnosis. While we did not find an article covering the exact examples mentioned by the reviewer, we have incorporated a very recent study from April 2023. This study assists in the differential diagnosis of Kaposi Sarcoma by detailing specific features that distinguish it from other similar cutaneous lesions using OCT.

4-Recognizing the reviewer's point that we extensively elaborated on OCT analysis in BCC, we have condensed that section. Additionally, following the reviewer's suggestion, in the subsequent paragraph (after BCC), we provide more detailed information on other skin lesions that may mimic KS.

Reviewer 2 Report

Authors present their experience utilizing OCT as an aid to KS diagnosis and its subsequent follow up in response to treatment. 

I agree that OCT may be a very useful complementary resource, particularly to investigate vascular changes in response to treatment. 

Minor comment: Please specify staining and magnification in figure 4c. 

Please verify that all wording is Ok. A few typos were detected. 

Author Response

We appreciate the reviewer's valuable feedback. Consequently, we have made the following corrections:

We have included the staining information and magnification in figure 4c as recommended by the reviewer.